# Association of Denture Use and Chewing Ability with Cognitive Function Analysed Using Panel Data from Korea Longitudinal Study of Aging (2006–2018)

**DOI:** 10.3390/healthcare11182505

**Published:** 2023-09-10

**Authors:** Nu-Ri Jun, Jae-Hyun Kim, Jong-Hwa Jang

**Affiliations:** 1Department of Public Health Science, Graduate School, Dankook University, Cheonan-si 31116, Republic of Korea; 12200639@dankook.ac.kr; 2Department of Health Administration, Dankook University, Cheonan-si 31116, Republic of Korea; jaehyun@dankook.ac.kr; 3Department of Dental Hygiene, College of Health Science, Dankook University, Cheonan-si 31116, Republic of Korea

**Keywords:** chewing ability, cognitive function, dental health, denture

## Abstract

This study aimed to investigate the association between denture use, chewing ability, and cognitive function in Korean middle-aged adults, using samples representing middle-aged people at the national level. This longitudinal study included 9998 middle-aged adults using the Korea Longitudinal Study of Aging 7th special survey data. Denture use, chewing ability, health-related factors, and general characteristics were assessed using the Computer-Assisted Personal Interview. After controlling for general characteristics using a generalized estimating equation model, the association of denture use and Mini-Mental State Examination (MMSE) scores with the chewing ability of those with or without dentures and MMSE scores were analysed. Among them, 24% of participants wore dentures, and 35.1% complained of difficulty chewing when wearing dentures. Among the participants who did not wear dentures, 16.4% complained of difficulty chewing. MMSE scores were lower among denture wearers than non-denture wearers (β = −0.026, *p* < 0.001). In both groups, MMSE scores decreased with chewing difficulty and were significantly reduced among non-denture wearers (*p* < 0.05). Chewing ability was closely associated with cognitive functioning. Therefore, in order to prevent cognitive decline, chewing discomfort while wearing dentures must be eliminated, and oral health must be maintained.

## 1. Introduction

People aged 65 years and older account for 17.5% of the current total population in Korea. Korea is projected to become a super-aged society, as older adults will account for 20.6% of its total population by 2025. [1]. The prevalence of cognitive function impairments such as dementia, a representative disease of older adults, is also increasing rapidly [2]. Dementia is caused by various disorders, including degenerative brain disorders, cerebrovascular disease, endocrine disease, hydrocephalus, and brain tumours; several recent studies report that oral health affects cognitive function [3,4]. Since oral health affects nutritional status in older adults, it should be considered together with physical and mental health [5]. Gingival bleeding, stomatitis, oral soft tissue disorders, such as mucosal lesions and decreased saliva volume, periodontal disease characterized by periodontal bone resorption, and tooth loss can be risk factors for cognitive decline [6,7,8].

The National Evidence-based Healthcare Collaborating Agency (NECA) defines oral frailty as ‘a decrease in physiological function due to a decline in oral and maxillofacial functions owing to aging,’ and reports an agreement on the diagnostic criteria and treatment methods for Koreans. Older adults aged ≥65 years are diagnosed with oral frailty if there is a decline in two or more of a total of six functions (chewing ability, bite force, tongue pressure, salivary gland function, swallowing function, and maintaining good oral hygiene) [9]. Among these, chewing ability, which is the first step in the serial digestion process of food, is closely associated with oral health [10], and partial or full tooth loss and decreased saliva due to aging can cause a decline in chewing ability in older adults [11,12].

One general method for overcoming decreased chewing ability is the use of partial or full dentures, and the number of people using dentures is increasing [13]. However, chewing ability does not improve immediately with denture wear because of difficulties in maintaining and adopting dentures due to issues such as a loss of oral mucous membrane elasticity, alveolar bone resorption, and xerostomia [14]. Further, several previous studies report that difficulty chewing is significantly associated with frailty [15,16,17]. Chewing increases activity not only in the hippocampus and prefrontal cortex, which play important roles in cognitive processes, but also in the primary somatosensory cortex (S1) and primary motor cortex (M1), and it has a positive influence on brain functions [18,19]. Therefore, decreased and impaired chewing ability reduces blood flow to the brain, induces chronic stress, inhibits spatial learning ability, and decreases cognitive function due to poor dietary intake [20].

Cognitive decline negatively impacts life and well-being among older adults, along with physical activity and function deterioration, reduced economic power, and isolation from social activity [21]. Although dementia causes impaired cognitive function, early detection and treatment can delay the aggravation of symptoms and improve cognitive decline through training [22]. The predementia phase is divided into mild cognitive impairment (MCI) and moderate cognitive impairment. MCI is a condition in which people experience more memory problems than their peers in terms of age and education level, despite their well-preserved performance of daily activities. Moderate cognitive impairment is a condition in which people are not able to recall recent events well and perform complex tasks efficiently and correctly [23]. The Mini-Mental State Examination (MMSE) is the most widely used screening instrument for dementia worldwide [24,25]. It is useful, relatively easy to apply, and can be administered easily.

Overall, we hypothesize that chewing difficulties may also affect cognitive decline and that chewing difficulties and cognitive decline are negatively related. Previous studies investigated the association between many chronic systemic diseases (diseases of the immune and cardiovascular systems) and cognitive function to understand the relevant factors for early detection of lesions [26,27]. The association between cognitive decline and both systemic health and oral health has drawn attention recently. Given this potential importance, this study investigated the association between chewing ability and dentures and cognitive function in middle-aged adults. Korea Longitudinal Study of Aging (KLoSA) is a large-scale longitudinal survey conducted at the national level and is considered to have high reliability and accuracy [28]. Therefore, this study used panel data from the KLoSA to investigate the association between denture use, cognitive function, and the association between chewing ability according to dentures and cognitive function.

## 2. Methods

### 2.1. Participants and Data Collection

This longitudinal survey study used panel data from the 7th KLoSA (2006–2018) [29]. The KLoSA is a cohort study conducted to provide basic data for various studies related to aging in community-dwelling Korean middle-aged adults (≥45 years) by the Korea Employment Information Service (KEIS) and is disclosed on the website as de-identified secondary data [30].

The KLoSA included 10,254 persons from 6171 households (1.7 persons per household) as panel respondents at the first baseline survey in 2006 who were followed up until death. The proportion of persons remaining in the panel at the 7th baseline survey was 78.8% [31]. Random, multistage, and stratified methods were used to select a probability sample by region and type of household. Then, systemic sampling was used to select samples. The survey has been conducted every 2 years. In even years, a baseline survey is conducted to investigate basic information that should be repeatedly measured. In odd years, employment-related specific information is selected, and a special survey is conducted to manage panel maintenance. This study included 9998 participants previously established using data from the 7th KLoSA special survey in 2019.

For data collection, trained investigators obtained voluntary consent to participate in the study from the participants and conducted a Computer Assisted Personal Interview (CAPI) based on the KLoSA standard protocol. Investigators for only the panel comprised senior investigators with at least 3 years of experience with KANTAR (KANTAR Group Ltd., London, UK), which was constructed in 2006 to maintain inter-investigator consistency of the collected data. To minimize the rate of panel dropouts, at least 80% of the investigators remained in the survey for 16 years since the first year. A standardized teaching plan for training contents and methods was used to educate the investigators after prior arrangements with KEIS. In the middle of the inspection, researchers of KEIS and KANTAR visited and supervised the investigators during survey administration and conducted a meeting to confirm their understanding of task-related knowledge, site information, and difficulties. They also inspected interim data collected within 1 month after the initiation of the survey to prevent abnormal values or logic errors from repeatedly occurring in advance and conducted a subsequent re-evaluation. After completing the survey via CAPI, response data entered in real-time were transferred by unit time to the KANTAR server for storage [30].

This study complied with the guidelines of the Declaration of Helsinki and was reviewed and approved by the Institutional Review Board of Dankook University Hospital (IRB No: DKU 2020-08-013).

### 2.2. Variables

This study included five items regarding denture use, chewing ability, MMSE, and sociodemographic characteristics and three items regarding health conditions and behavioural factors.

#### 2.2.1. Independent Variables

Wearing dentures was defined as wearing all types of dentures, regardless of whether complete or partial. Denture use was evaluated by the question ‘Do you usually wear dentures?’ and answered by ‘yes’ or ‘no.’ The question ‘Can you take bites out of or chew hard food, such as apples or meats with no effort when you are wearing dentures?’ was used to assess the chewing ability regarding denture use. Answer options for the question were ‘chewing very well,’ ‘chewing well,’ ‘moderate,’ ‘inability to chew well,’ and ‘not chewing at all.’ Chewing ability while not wearing dentures was assessed by the question ‘Can you take bites out of or chew hard food such as apples or meats with no effort without wearing dentures?’ Answer options were ‘chewing very well,’ ‘chewing well,’ ‘moderate,’ ‘inability to chew well,’ and ‘not chewing at all’.

#### 2.2.2. Dependent Variables

Cognitive function was assessed using the Korean version of the MMSE (K-MMSE) for dementia screening [25]. The K-MMSE assesses five categories: orientation to time (5 points), orientation to place (5 points), recall (3 points), memory registration (3 points), attention and calculation (5 points), and language and visuospatial abilities (9 points). Higher scores indicate better cognitive performance. A score of 23–24 out of a total score of 30 points is considered the cut-off indicative of cognitive impairment, while a score of ≥24 is considered ‘normal’, and scores of 18–23 and ≤17 are considered ‘MCI’ and ‘suspected dementia’, respectively.

#### 2.2.3. Control Variables

Sociodemographic characteristics (age, education level, sex, marital status, and working restriction), health conditions, and behavioural factors (alcohol consumption, health insurance type, and the number of diseases) were included as covariates [32]. Participants were divided by age groups: ≤54 years, 55–64 years, 65–74 years, and ≥75 years, and by education level: ≤elementary school, middle school, high school, and ≥college. Sex was categorized as either male or female. Marital status was categorized into married, divorced/separated, and single.

Working restriction due to health conditions was assessed by answering ‘yes’ or ‘no.’ In the health condition and behavioural factors, alcohol consumption was assessed by answering ‘yes’ or ‘no,’ and health insurance was categorized into national health insurance and medical aid. Regarding chronic diseases, the number of diseases, such as hypertension, diabetes, cancer, chronic obstructive pulmonary disease, liver disease, cardiovascular disease, cerebrovascular disease, mental disease, and arthritis, were summed and classified into “0,” “1,” or “≥2.”

### 2.3. Statistical Analyses

Statistical analyses were conducted using SAS software (Version 9.4; SAS Institute Inc., Cary, NC, USA). The independent *t*-test and analysis of variance (ANOVA) were used to compare changes in MMSE scores according to denture use and chewing ability by general characteristics. The Generalized Estimating Equation (GEE) model was used to control all general characteristics. The association between MMSE levels by denture use, MMSE levels by the chewing ability of denture wearers, and MMSE levels by the chewing ability of non-denture wearers was analysed. Significance was set at *p* < 0.05.

## 3. Results

### 3.1. General Characteristics

Participants’ sociodemographic characteristics are shown in Table 1. Out of 9998 participants, 2402 (24.0%) wore dentures and 7596 (76.0%) did not. Cognitive function scores for denture wearers (22.96 ± 6.2) were lower than non-denture wearers (26.22 ± 4.7) (*p* < 0.001).

Among denture wearers, 36.4% and 32.7% answered the question about chewing ability with ‘moderate’ and ‘inability to chew well,’, respectively. The lowest mean MMSE score (18.50 ± 7.4) was observed for participants who could not chew at all, indicating that MMSE scores were significantly negatively associated with chewing ability (*p* < 0.001). Among non-denture wearers, 51.2% and 26.0% answered the question about chewing ability with ‘chewing very well’ and ‘moderate,’, respectively. The lowest mean MMSE score (18.39 ± 7.7) was observed in those who could not chew at all, indicating that the MMSE scores were significantly negatively associated with chewing ability, similar to the results of those with dentures (*p* < 0.001).

Regarding age distribution, 32.4%, 27.4%, 26.3%, and 13.9% were ≤54 years, 55–64 years, 65–74 years, and ≥65 years, respectively. Most participants (46.8%) had an elementary education or higher, and more participants were female (56.4%) than male (43.6%). Most participants were married (78.2%), and 66.4% and 62.5% of the participants answered ‘no’ to questions regarding working restrictions and alcohol consumption, respectively. Regarding health insurance, most participants (93.8%) were under the national health insurance program, and 51.9% had no chronic diseases, whereas 28.9% and 19.2% had one disease and at least two diseases, respectively.

The MMSE scores decreased significantly as age increased and the education level decreased (*p* < 0.001). MMSE scores were significantly lower for participants who were female, divorced/separated, under medical aid, and with working restrictions (*p* < 0.001). Although MMSE scores were not significantly associated with alcohol consumption (*p* = 0.013) or number of chronic diseases (*p* = 0.274), they were lower for drinkers than non-drinkers, and lower for those with at least two chronic diseases than those with no disease or one disease.

### 3.2. Association between Denture Use and MMSE

Table 2 shows the results of the analysis of the association between denture use and MMSE scores after adjustment for control variables.

MMSE scores of denture wearers were 0.026 points (95% confidence interval [CI] = −0.030 to −0.022, *p* < 0.001) lower than those of non-denture wearers. In terms of age, MMSE scores among older adults aged ≥75 years were 0.186 points (95% CI = −0.192 to −0.179, *p* < 0.001) lower than those of adults aged ≤ 54 years, establishing a negative correlation indicating that as age increased, cognitive function significantly decreased. In terms of education, the MMSE scores of participants with elementary education or lower were 0.098 points (95% CI = −0.104 to −0.093, *p* < 0.001) lower than those of participants who were college graduates, indicating that lower education levels are associated with lower cognitive function. According to the health insurance type, the MMSE scores of participants with medical aid were 0.040 points (95% CI = −0.048 to −0.033, *p* < 0.001) lower than those of participants with national health insurance. Participants with working restrictions had MMSE scores that were 0.062 points (95% CI = −0.066 to −0.059 *p* < 0.001) lower than those of participants with no working restrictions. Compared to 2018, the MMSE scores of all the participants were −0.016 points (95% CI = −0.022 to −0.010, *p* < 0.001), −0.022 points (95% CI = −0.027 to −0.016, *p* < 0.001), and −0.020 points (95% CI = −0.026 to −0.014, *p* < 0.001) lower in 2006, 2008, and 2010, respectively.

### 3.3. Association between MMSE and Chewing Ability among Denture Wearers and Non-Denture Wearers

Table 3 shows the results of the analysis of the association between MMSE scores and the chewing ability of denture wearers versus non-denture wearers after adjustment for other control variables.

Among denture wearers (Model 1), compared to the participants who answered ‘chewing very well’ for the question about chewing ability, the MMSE scores for participants who answered ‘inability to chew well’ and ‘not chewing at all’ were 0.080 points (95% CI = −0.126 to −0.035, *p* < 0.001) and 0.143 points (95% CI = −0.200 to −0.086, *p* < 0.001) lower, respectively. The MMSE scores of participants with dentures in 2006 were −0.020 points (95% CI = −0.037 to −0.002, *p* = 0.026) lower than those in 2018.

Among non-denture wearers (Model 2), compared to the participants who answered ‘chewing very well’ for the question about chewing ability, the MMSE scores for participants who answered ‘inability to chew well’ and ‘not chewing at all’ were 0.079 points (95% CI = −0.088 to −0.071, *p* < 0.001) and 0.220 points (95% CI = −0.241 to −0.199, *p* < 0.001) lower, respectively. In Model 2, as chewing ability decreased, cognitive function was significantly reduced (*p* < 0.05). Compared to 2018, the MMSE scores of all participants in 2006, 2008, and 2010 were −0.013 points (95% CI = −0.019 to −0.007, *p* < 0.001), −0.020 points (95% CI = −0.025 to −0.014, *p* < 0.001), and −0.019 points (95% CI = −0.025 to −0.013, *p* < 0.001) lower, respectively.

## 4. Discussion

Cognitive impairment, an age-related disease, is drawing social attention commensurate with the rapid increase in population age [24]. This study analysed the association of denture use and chewing ability with cognitive function among middle-aged adults using data from the KLoSA. KLoSA, conducted by KEIS, investigated changes in cognitive function scores using data from longitudinal studies conducted from 2006 to 2018 [28,29,30,31].

This study aimed to demonstrate the mechanism between chewing ability affecting oral health conditions, the use of dentures somewhat supplementing decreased chewing ability due to tooth loss, and cognitive function.

Cognitive function was assessed using the K-MMSE [24]. After adjustment of various variables in this study, we observed that MMSE scores were significantly associated with age, education level, sex, marital status, working restriction, and health insurance. Previous evidence indicates that among sociodemographic characteristics, education level and age are most largely attributable to MMSE score prediction [33,34,35]. The MMSE scores decreased as education level decreased and age increased [34]. This shows that age and education level can be considered in the development of standard data for MMSE. Also, compared to men, women had lower MMSE scores, which is consistent with previous evidence that cognitive decline is significantly much faster in women than men [35]. This is also supported by a survey indicating that the incidence and prevalence of dementia are much higher in women than men [2]. Additionally, participants under medical aid had lower MMSE scores than those under national health insurance. One study reported that health literacy significantly affects cognitive function [36], which is consistent with previous evidence that total health literacy scores for patients under medical aid were significantly lower than for patients under national health insurance [37].

After adjustment for all variables, in this study, the association between MMSE scores and denture use was analysed and we observed that MMSE scores of denture wearers were 0.026 points lower than those of non-denture wearers. Wearing dentures can restore the function of missing teeth and reduce cognitive decline progression [38]. However, since the oral sensory functions of denture wearers were more deteriorated compared to those with original teeth, denture wearers showed lower cognitive function scores than non-denture wearers [14]. This result was similar to a previous report that 90% of participants with inadequate chewing ability who wore partial dentures had a greater risk of dementia than those with natural masticatory function [39]. Denture wearers have fewer remaining teeth, and the number of teeth also affects cognitive function [40]. Since cognitive decline in patients with Alzheimer’s disease aggravates dental care and causes an increase in mucosal lesions, such as denture stomatitis, not only denture use but also the necessity of appropriate management is emphasized [41].

This study analysed the association between cognitive function and chewing ability by using the GEE. In both groups (denture wearers and non-denture wearers), compared to the participants who answered ‘chewing very well,’ the MMSE scores decreased for participants who answered ‘inability to chew well’ and ‘not chewing at all.’ This reduction was significant among non-denture wearers. Accordingly, chewing ability affected cognitive function and significantly decreased cognitive function was observed among participants with chewing difficulties. According to a study by Takehara et al., older male adults who had <20 natural teeth with limited chewing ability were more likely to have cognitive impairment [42]. This is consistent with our results indicating that chewing ability and cognitive function are proportionally associated. Further, a study evaluating chewing ability, functional elements, and diet in community-dwelling older adults revealed that decreased chewing ability is associated with not only cognitive function but also poor activities of daily living (ADL), depression, and dietary deficiency [43]. Data from the 6th Korea National Health and Nutrition Examination Survey were analysed, revealing that 61.7% of older adults aged ≥65 years complained of chewing difficulties. In other words, more than half of older adults complain of chewing difficulties [44]. Chewing difficulty narrows the range of food options, resulting in poor diet quality and nutritional imbalance, which also increases the prevalence of systemic diseases and reduces health-related quality of life [45]. Moreover, having fewer teeth is associated with poor performance in activities of daily living [46]. Further, poor oral health with chewing difficulty is a risk factor for mortality among Korean middle-aged adults who exercise regularly [32].

This suggests the necessity for a program that can prevent cognitive impairment and dementia by enhancing chewing ability, preventing depression, and promoting exercise ability. In addition to the above, correct mastication is also considered important, along with the number of teeth, method of chewing, and degree of crushing. In particular, one-sided chewing can cause dental attrition, periodontal disease, and temporomandibular joint disorders [47]. Chewing using molars can have more significant impacts on the stimulation of cognitive function because greater relative molar occlusal balance was associated with increased cognitive function in older adults [48]. The clinical significance of this study was that denture users had low cognitive function; however, there was no correlation with cognitive function when there was little or no chewing discomfort even when wearing dentures. This suggests that even if the number of teeth is small, cognitive function may not decline if chewing discomfort is resolved by wearing dentures suitable for one’s oral cavity. Therefore, reduced chewing ability is a risk factor for cognitive decline and dementia. Maintaining good chewing ability and resolving poor chewing ability may be important for preventing dementia. Therefore, it is necessary to investigate several specific factors related to mastication and identify the mechanisms directly affecting cognitive function. Additionally, further studies are needed to demonstrate the effectiveness of continuous oral health management and professional oral muscle function training to delay decreased chewing ability in people with impaired cognitive function.

### Limitations and Future Research

Various studies on the association between systemic factors and cognitive function have been discussed. However, this study is noteworthy, given the dearth of Korean longitudinal studies analysing the association between denture use, oral health-related factors, and chewing ability.

Since this survey study used and analysed secondary data, it has some limitations. First, since this study did not conduct oral examinations using direct investigation, data were obtained from interview surveys. Chewing ability, which can vary depending on the type of dentures (full or partial dentures), was not considered in assessing denture wear. For non-denture wearers, accurate oral health condition was not determined. Further studies are required to present additional items for denture types and consider objective indicators demonstrating oral health conditions, such as remaining teeth.

The second limitation is that among the investigated items, masticatory function was measured with a subjective assessment of chewing (hard food such as apples or meats) ability. According to the 2nd National Oral Health Plan for 5 years, recently presented by the Ministry of Health and Welfare (MOHW), the MOHW will review the introduction of masticatory function tests in national screening [49]. In future large-scale studies, such as national screening, we expect that more multilateral studies can be conducted by constructing data based on measurement and evaluation of chewing ability using objective indicators.

## 5. Conclusions

A significant association between denture use, chewing ability, and cognitive function was observed among Korean middle-aged adults. Difficulty chewing is associated with cognitive decline. This is consistent with previous findings that cognitive function decreases with denture use and greater chewing difficulties. MMSE scores were lower among participants with difficulty chewing, regardless of denture use. Particularly, when chewing discomfort was less than moderate even with the use of dentures, there was no significant relationship with cognitive function. It can be inferred that wearing dentures suitable for one’s oral cavity can prevent cognitive decline. Therefore, active oral health-promoting behaviours, such as chewing training that can increase chewing ability, may be appropriate for preserving cognitive function and should be considered for dementia prevention programs to preserve cognitive function. To achieve this, an intervention study is necessary to analyse the effectiveness of oral muscle function-enhancing training that can increase chewing ability, and hence, perhaps, cognitive function. Moreover, further studies are required to investigate the efficacy of various programs for maintaining good chewing ability and the early prevention of cognitive impairment based on community-based public medical centres.

## Figures and Tables

**Table 1 healthcare-11-02505-t001:** General characteristics of participants included for analysis at baseline (2006).

	Total		MMSE		
	n	%	Mean	SD	*p*-Value
Whether you usually wear dentures					<0.001
Yes	2402	24.0	22.96	6.2	
No	7596	76.0	26.22	4.7	
Chewing hard food when wearing a denture					<0.001
Chewing very well	30	1.3	25.03	4.6	
Chewing well	654	27.2	24.20	5.4	
Usually	874	36.4	23.58	5.8	
Inability to chew well	786	32.7	21.50	6.8	
Not chewing at all	58	2.4	18.50	7.4	
Chewing hard foods without the usual denture					<0.001
Chewing very well	483	6.4	27.50	3.5	
Chewing well	3890	51.2	27.45	3.4	
Usually	1978	26.0	25.41	4.9	
Inability to chew well	1129	14.9	23.63	6.2	
Not chewing at all	116	1.5	18.39	7.7	
Age					<0.001
≤54	3238	32.4	28.04	2.6	
55–64	2742	27.4	26.56	3.8	
65–74	2633	26.3	24.21	5.2	
≥75	1385	13.9	19.46	7.2	
Education level					<0.001
Elementary school or less	4678	46.8	22.80	6.1	
Middle school	1628	16.3	27.06	3.3	
High school	2662	26.6	27.91	3.0	
College or higher	1030	10.3	28.49	2.4	
Sex					<0.001
Male	4359	43.6	26.65	4.3	
Female	5639	56.4	24.50	5.8	
Marital status					<0.001
Married	7813	78.2	26.35	4.4	
Separated, divorced	2101	21.0	21.99	6.8	
Single	84	0.8	26.63	5.3	
Working restriction					<0.001
Yes	3363	33.6	22.81	6.5	
No	6635	66.4	26.77	4.0	
Alcohol consumption					0.013
Yes	3752	37.5	26.76	4.0	
No	6246	62.5	24.64	5.8	
Health insurance					<0.001
NHI	9377	93.8	25.64	5.2	
Medical aid	621	6.2	22.43	6.5	
Number of chronic diseases *					0.274
0	5184	51.9	26.49	4.6	
1	2890	28.9	24.86	5.5	
≥2	1924	19.2	23.47	6.1	
Total	9998	100.0	25.44	5.3	

* Hypertension, diabetes, cancer, chronic obstructive pulmonary disease, liver disease, cardiovascular disease, cerebrovascular disease, and arthritis; *p*-values were calculated using the independent *t*-test or one-way analysis of variance (ANOVA) test at α = 0.01; MMSE, Mini-Mental State Examination; NHI, national health insurance; SD, standard deviation.

**Table 2 healthcare-11-02505-t002:** Association between denture use and MMSE scores.

	MMSE			
	*B*	95% CI		*p*-Value
Whether you usually wear dentures				
Yes	−0.026	−0.030	−0.022	<0.001
No	ref			
Age				
≤54	ref			
55–64	−0.013	−0.017	−0.009	<0.001
65–74	−0.050	−0.054	−0.045	<0.001
≥75	−0.186	−0.192	−0.179	<0.001
Education level				
Elementary school or less	−0.098	−0.104	−0.093	<0.001
Middle school	−0.025	−0.030	−0.019	<0.001
High school	−0.013	−0.018	−0.008	<0.001
College or higher	ref			
Sex				
Male	ref			
Female	−0.019	−0.023	−0.016	<0.001
Marital status				
Married	ref			
Separated, divorced	−0.044	−0.048	−0.040	<0.001
Single	−0.030	−0.047	−0.014	<0.001
Working restriction				
Yes	−0.062	−0.066	−0.059	<0.001
No	ref			
Alcohol consumption				
Yes	ref			
No	−0.016	−0.019	−0.012	<0.001
Health insurance				
NHI	ref			
Medical aid	−0.040	−0.048	−0.033	<0.001
Number of chronic diseases *				
0	ref			
1	−0.001	−0.006	0.003	0.586
≥2	−0.018	−0.026	−0.010	<0.001
Year				
2006	−0.016	−0.022	−0.010	<0.001
2008	−0.022	−0.027	−0.016	<0.001
2010	−0.020	−0.026	−0.014	<0.001
2012	−0.004	−0.010	0.002	0.175
2014	−0.005	−0.011	0.001	0.077
2016	0.002	−0.004	0.008	0.457
2018	ref			

*p*-values were calculated using a Generalized Estimating Equation (GEE) model at α = 0.01. The model was adjusted for all other variables except the target variable. * Hypertension, diabetes, cancer, chronic obstructive pulmonary disease, liver disease, cardiovascular disease, cerebrovascular disease, and arthritis; MMSE, Mini-Mental State Examination; CI, confidence interval; NHI, national health insurance.

**Table 3 healthcare-11-02505-t003:** Association between MMSE scores and chewing ability of denture wearers and non-denture wearers.

	MMSE							
	*B*	95% CI		*p*-Value	*B*	95% CI		*p*-Value
	Model 1				Model 2			
Chewing hard food when wearing a denture								
Chewing very well	ref							
Chewing well	−0.006	−0.052	0.040	0.794				
Usually	−0.005	−0.050	0.041	0.841				
Inability to chew well	−0.080	−0.126	−0.035	0.001				
Not chewing at all	−0.143	−0.200	−0.086	<0.001				
Chewing hard foods without the usual denture								
Chewing very well					ref			
Chewing well					0.008	0.001	0.015	0.033
Usually					−0.018	−0.025	−0.010	<0.001
Inability to chew well					−0.079	−0.088	−0.071	<0.001
Not chewing at all					−0.220	−0.241	−0.199	<0.001
Age								
≤54	ref				ref			
55–64	−0.020	−0.047	0.008	0.164	−0.011	−0.015	−0.007	<0.001
65–74	−0.062	−0.089	−0.035	<0.001	−0.042	−0.047	−0.038	<0.001
≥75	−0.189	−0.216	−0.161	<0.001	−0.160	−0.167	−0.154	<0.001
Education level								
Elementary school or less	−0.103	−0.123	−0.082	<0.001	−0.089	−0.094	−0.084	<0.001
Middle school	−0.011	−0.033	0.011	0.321	−0.028	−0.033	−0.023	<0.001
High school	−0.015	−0.037	0.006	0.158	−0.015	−0.019	−0.010	<0.001
College or higher	ref				ref			
Sex								
Male	ref				ref			
Female	−0.048	−0.058	−0.037	<0.001	−0.012	−0.016	−0.009	<0.001
Marital status								
Married	ref				ref			
Separated, divorced	−0.061	−0.072	−0.050	<0.001	−0.029	−0.033	−0.025	<0.001
Single	0.026	−0.035	0.086	0.409	−0.035	−0.050	−0.019	<0.001
Working restriction								
Yes	−0.087	−0.097	−0.078	<0.001	−0.041	−0.045	−0.038	<0.001
No	ref				ref			
Alcohol consumption								
Yes	ref				ref			
No	−0.025	−0.035	−0.015	<0.001	−0.015	−0.018	−0.011	<0.001
Health insurance								
NHI	ref				ref			
Medical aid	−0.037	−0.054	−0.019	<0.001	−0.032	−0.040	−0.024	<0.001
Number of chronic diseases *								
0	ref				ref			
1	0.001	−0.010	0.013	0.831	0.000	−0.005	0.004	0.886
≥2	0.001	−0.017	0.020	0.905	−0.019	−0.027	−0.010	<0.001
Year								
2006	−0.020	−0.037	−0.002	0.026	−0.013	−0.019	−0.007	<0.001
2008	−0.009	−0.025	0.008	0.315	−0.020	−0.025	−0.014	<0.001
2010	−0.017	−0.034	0.000	0.050	−0.019	−0.025	−0.013	<0.001
2012	0.000	−0.017	0.016	0.962	−0.006	−0.012	0.000	0.062
2014	−0.020	−0.038	−0.002	0.026	−0.002	−0.008	0.003	0.432
2016	0.012	−0.005	0.030	0.171	0.000	−0.006	0.006	0.978
2018	ref							

*p*-values were calculated using a Generalized Estimating Equation (GEE) model at α = 0.01. All models were adjusted for all other variables except the target variable; Model 1 was adjusted for all variables among chewing hard foods when wearing a denture; Model 2 was adjusted for all variables among chewing hard food without the usual denture. * Hypertension, diabetes, cancer, chronic obstructive pulmonary disease, liver disease, cardiovascular disease, cerebrovascular disease, and arthritis; MMSE, Mini-Mental State Examination; CI, confidence interval; NHI, national health insurance.

## Data Availability

The data of the KLoSA are publicly available on the KLoSA website (https://survey.keis.or.kr/klosa/klosa01.jsp (accessed on 20 May 2023). The data presented in this study are available on request from the corresponding author.

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
