# Peer review of "Association of Denture Use and Chewing Ability with Cognitive Function Analysed Using Panel Data from Korea Longitudinal Study of Aging (2006–2018)"

_healthcare, 2023, doi:10.3390/healthcare11182505_

Round 1

Reviewer 1 Report

The manuscript is very interesting.

However, minor issues need to be considered:

1- The authors need to specify which type of denture (line 2) partial or complete.

2- Line 14-15. The abstract start with a sentence that needs evidence. The sentence should be removed or adjusted.

3- Many Abbreviations used in the manuscript need to be expanded and explained after the first use.

Example:

Line 22 (MMSE).,  Line 123 (KANTAR), Line 344 (ADL).

4- The association does not imply causation, Line 29-30. Therefore, the sentence in lines 29-30 should be rewritten.

5- Line 90 91: The sentence is not clear.

6- Line 319-320 The sentence needs a reference (evidence).

7- Line 359-360 The sentence needs a reference (evidence)

8- Line 367 369: The sentence is not clear.

9- Line 406: Using the verb (improve) doesn’t match the study results or the references used. Maybe using the verb (preserve) is more suitable?

Line 90 -91: The sentence is not clear.

Line 367- 369: The sentence is not clear.

Line 406: Using the verb (improve) doesn’t match the study results or the references used. Maybe using the verb (preserve) is more suitable?

Author Response

The manuscript is very interesting. However, minor issues need to be considered.

Thank you for reviewing our research. Below are our responses to your comments and queries. We have tried to reflect your valuable comments in our revisions as much as possible.

  1. The authors need to specify which type of denture (line 2) partial or complete.

Authors’ response:

We appreciate your valuable suggestion. In our study, denture wearing was defined as wearing all types of dentures, including both complete and partial dentures, since there was no differentiation between partial and complete dentures in the panel study. In response to the reviewer's comments, an operational definition of denture wearing was added in the Methods section as follows:

  • “Wearing dentures was defined as wearing all types of dentures, regardless of whether complete or partial.”
  1. Line 14-15. The abstract start with a sentence that needs evidence. The sentence should be removed or adjusted.

Authors’ response:

We appreciate your valuable suggestion and agree with your point. Thus, we decided to delete the following, “Very poor oral health, common among older adults, is associated with cognitive decline”.

  1. Many Abbreviations used in the manuscript need to be expanded and explained after the first use. Example : Line 22 (MMSE), Line 123 (KANTAR), Line 344 (ADL).

Authors’ response:

We appreciate your valuable suggestion. However, KANTAR is the world’s leading data, insights, and consulting company. Thus, it is not an abbreviation. Nevertheless, the terms have been revised as follows:

  • “Mini-mental State Examination (MMSE), activities of daily living (ADL), KANTAR (KANTAR Group Ltd., London, England, UK) ”
  1. The association does not imply causation, Line 29-30. Therefore, the sentence in lines 29-30 should be rewritten.

Authors’ response:

Thank you for your valuable comment. In response to this, we have revised the content as follows:

  • “Therefore, in order to prevent cognitive decline, chewing discomfort while wearing dentures must be eliminated, and maintaining oral health must be maintained.”
  1. Line 90- 91: The sentence is not clear.

Authors’ response:

Thank you for your valuable comment. In response to this, we have revised the content as follows:

  • “…and that chewing difficulties and cognitive decline are negatively related. ”
  1. Line 319-320 The sentence needs a reference (evidence).

Authors’ response:

Thank you for pointing this out. We have added the following reference.

  • “[38] Yang, H.L.; Li, F.R.; Chen, P.L.; Cheng, X.; Mao, C.; Wu, X.B. (2022). Tooth loss, denture use, and cognitive impairment in Chinese older adults: a community cohort study. J Gerontol A 2022, 77, 180-187.”

  1. Line 359-360 The sentence needs a reference (evidence).

Authors’ response:

Thank you for pointing this out. We have added the following reference:

  • “[47] Isnaeni, R.S.; Patria, A.; Silvana, I.R. Relationship of one side chewing habits to temporomandibular joint disorders occurrence. Med. Dent. Sci. 2022, 2, 279-302.
  1. Line 367- 369: The sentence is not clear.

Authors’ response:

Thank you for your valuable comment. Thus, we have decided to delete the following sentence:

  • “Chewing disability can cause anatomical problems, such as oral soft tissues (oral mucous membrane and tongue), jaw joint or chewing muscles around the jaw joint [49].”
  1. Line 406: Using the verb (improve) doesn’t match the study results or the references used. Maybe using the verb (preserve) is more suitable?

Authors’ response:

Thank you for your valuable comment. Thus, we have revised the verb as follows:

  • “Therefore, active oral health-promoting behaviours, such as chewing training, that can increase chewing ability may be appropriate for preserving cognitive function and should be considered for dementia prevention programs to preserve cognitive function.”

We have made our best efforts to accommodate your recommendations in the revised manuscript. Please let us know in detail if you have any further recommendations for modifications. We would be glad to incorporate any required further revisions. Thank you very much.

Reviewer 2 Report

This is a very interesting survey to find a relationship between denture use and chewing ability and cognitive function. However, additional information is required to improve the clarity of the paper :

1. What are the age range of the denture wearers and non denture wearers? Does the mean age of denture wearers higher than denture wearers?

2. For the non denture wearers - are they not wearing denture because they have enough natural teeth to eat or are they not wearing dentures because their dentures are not comfortable to wear?

3. What is chewing training? Is this program/training really necessary? How about ensuring the edentulous or partially edentulous elderly are provided with good quality functional denture that they can eat with? Perhaps training of the dentists in providing good quality dentures are more useful. 

Author Response

This is a very interesting survey to find a relationship between denture use and chewing ability and cognitive function. However, additional information is required to improve the clarity of the paper :

Thank you for reviewing our research. Below are our responses to your comments and queries. We have tried to reflect your valuable comments in our revisions as much as possible.

  1. What are the age range of the denture wearers and non denture wearers? Does the mean age of denture wearers higher than denture wearers?

Authors’ response:

Thank you for your thoughtful comments. The participants in our study were middle-aged and older than 45 years old, and the age range was less than 54 years old (32.4%), 55–64 years old (27.4%), 65–74 years old (26.3%), and 75 years old or older (13.9%). In the GEE model, in which the relationship between denture use, chewing discomfort, and cognitive function was analyzed, age was analyzed as a control variable to prevent confusion that may be caused by age.

  1. For the non denture wearers - are they not wearing denture because they have enough natural teeth to eat or are they not wearing dentures because their dentures are not comfortable to wear?

Authors’ response:

Thank you for your inquisitive question. In general, people who do not wear dentures may have a large number of remaining natural teeth. Therefore, in our study, we focused on analyzing the relationship between chewing discomfort and cognitive function depending on whether or not dentures were worn. In general, cognitive function is reportedly lowered when the number of teeth is small; however, if dentures are necessary from a clinical point of view, cognitive function may not be affected if chewing discomfort is relieved by wearing dentures suitable for one's mouth. These findings have been added to the Discussion and Conclusions sections.

  1. What is chewing training? Is this program/training really necessary? How about ensuring the edentulous or partially edentulous elderly are provided with good quality functional denture that they can eat with? Perhaps training of the dentists in providing good quality dentures are more useful.

Thank you for your valuable comment. Below are our responses to your comments

Authors’ response:

Thank you very much for your constructive comments. In general, chewing discomfort may be felt with decreased number of teeth as age increases. Aside from that, periodontal disease may occur or the muscles around the mouth may be weakened, resulting in lower masticatory ability. Thus, people who do not have teeth must wear dentures that are suitable for their mouth. However, irrespective of whether dentures are worn or not, masticatory and swallowing functions can be improved through oral muscle function training. In addition to supplying high-quality dentures, oral muscle function training can improve not only masticatory function but also oral health for middle-aged and older adults. This as further described in the Discussion and Results sections as follows:

  • Discussion section: The clinical significance of this study was that denture users had low cognitive function; however, there was no correlation with cognitive function when there was little or no chewing discomfort even when wearing dentures. This suggests that even if the number of teeth is small, cognitive function may not decline if chewing discomfort is resolved by wearing dentures suitable for one's oral cavity.
  • Conclusion section: Particularly, when chewing discomfort was less than moderate even with the use of dentures, there was no significant relationship with cognitive function. It can be inferred that wearing dentures suitable for one's oral cavity can prevent cognitive decline.

We have made our best efforts to accommodate your recommendations in the revised manuscript. Please let us know in detail if you have any further recommendations for modifications. We would be glad to incorporate any required further revisions. Thank you very much.

Reviewer 3 Report

The authors investigated the associations between denture use and chewing ability with cognitive function. I think this study has severe problems for publication.

             This study investigated association of denture use and cognitive function, and showed that cognitive function is lower in denture wearers than the non-denture wearers. However, the denture use itself did not have bad effects on cognitive function. As like authors said (line 320-), the cofounding factors of denture use, such as less teeth and low oral functions, would be associated with cognitive function. Thus, it is nonsense to investigate the association of cognitive function and denture use.

              In this study, the need for dentures of each participants was not evaluated. If the participants is limited in the only patient who needs dentures objectively, I think the results may be converse. That is, the denture users has better cognitive function.

Author Response

The authors investigated the associations between denture use and chewing ability with cognitive function. I think this study has severe problems for publication.

Thank you for reviewing our research. Below are our responses to your comments and queries. We have tried to reflect your valuable comments in our revisions as much as possible.

             This study investigated association of denture use and cognitive function, and showed that cognitive function is lower in denture wearers than the non-denture wearers. However, the denture use itself did not have bad effects on cognitive function. As like authors said (line 320-), the cofounding factors of denture use, such as less teeth and low oral functions, would be associated with cognitive function. Thus, it is nonsense to investigate the association of cognitive function and denture use.

Authors’ response:

Thank you for your thoughtful comments. The main purpose of our study is to confirm the association between the degree of chewing discomfort and cognitive function according to the use of dentures. In our study results, we were able to confirm the grounded theory that the low cognitive function in denture users was due to the decline in cognitive function because of the small number of teeth. However, even in denture wearers, there was no significant association with cognitive function if there was little or no chewing discomfort, but there was a significant correlation between masticatory discomfort or severe chewing discomfort and cognitive function. This means that even if the number of teeth is small, cognitive decline can be prevented if chewing discomfort is resolved. Interpretation of these results was further emphasized in the Discussion and Conclusions section as follows:

  • Discussion section: The clinical significance of this study was that denture users had low cognitive function; however, there was no correlation with cognitive function when there was little or no chewing discomfort even when wearing dentures. This suggests that even if the number of teeth is small, cognitive function may not decline if chewing discomfort is resolved by wearing dentures suitable for one's oral cavity.
  • Conclusion section: Particularly, when chewing discomfort was less than moderate even with the use of dentures, there was no significant relationship with cognitive function. It can be inferred that wearing dentures suitable for one's oral cavity can prevent cognitive decline.

              In this study, the need for dentures of each participants was not evaluated. If the participants is limited in the only patient who needs dentures objectively, I think the results may be converse. That is, the denture users has better cognitive function.

Authors’ response: 

Thank you for your meaningful comments. Unfortunately, our study did not assess each participant's need for dentures. However, one clinical significance of our study was that there was no association with cognitive function when there was no chewing discomfort due to denture wearing, and only when chewing discomfort was still unresolved. It can be inferred that the use of dentures suitable for one's oral cavity does not result in cognitive decline. We have emphasized this fact in the consideration and conclusions.

We have made our best efforts to accommodate your recommendations in the revised manuscript. Please let us know in detail if you have any further recommendations for modifications. We would be glad to incorporate any required further revisions. Thank you very much.

Round 2

Reviewer 2 Report

The rebuttal and revision is satisfactory. Thank you.

Reviewer 3 Report

I understood what you said.

To easy understanding, it is better that "model 1" and "model 2" in Table 3 are changed into such as "denture wearer/user" and "non-denture wearer/user" respectively.

But, it is OK you choice what you like.